# Healthy Food Environments in Early Learning Services: An Analysis of Manager Survey Responses, Menus and Policies in Regional New Zealand Early Childhood Education and Care Centres

**DOI:** 10.3390/ijerph19084709

**Published:** 2022-04-13

**Authors:** Pippa McKelvie-Sebileau, Erica D’Souza, David Tipene-Leach, Boyd Swinburn, Sarah Gerritsen

**Affiliations:** 1School of Population Health, University of Auckland, Auckland 1145, New Zealand; boyd.swinburn@auckland.ac.nz (B.S.); s.gerritsen@auckland.ac.nz (S.G.); 2Research and Innovation Centre, Eastern Institute of Technology, Napier 4112, New Zealand; dtipene-leach@eit.ac.nz; 3School of Future Environments, AUT University, Auckland 1010, New Zealand; e.dsouza@aut.ac.nz

**Keywords:** childcare centres, early childhood education and care, child welfare, Child Day Care Centers, food environments, early childhood, nutrition, regional New Zealand

## Abstract

Healthy food environments in early childhood play an important role in establishing health-promoting nutritional behaviours for later life. We surveyed Early Learning Services (ELS) in the Hawke’s Bay region of New Zealand and describe common barriers and facilitators to providing a healthy food environment, through descriptive survey analysis and thematic analysis of open-ended questions. We used a policy analysis tool to assess the strength and comprehensiveness of the individual centre’s nutrition policies and we report on the healthiness of menus provided daily in the centres. Sixty-two centres participated and 96.7% had policies on nutrition compared to 86.7% with policies on drinks. Of the 14 full policies provided for analysis, identified strengths were providing timelines for review and encouraging role modelling by teachers. The main weaknesses were communication with parents and staff, lack of nutrition training for staff and absence of policies for special occasion and fundraising food. With regard to practices in the ELS, food for celebrations was more likely to be healthy when provided by the centre rather than brought from home. Food used in fundraising was more likely to be unhealthy than healthy, though <20% of centres reported using food in fundraising. Only 40% of menus analysed met the national guidelines by not including any ‘red’ (unhealthy) items. Centre Managers considered the biggest barriers to improving food environments to be a lack of parental support and concerns about food-related choking. These results highlight the need for future focus in three areas: policies for water and milk-only, celebration and fundraising food; increased nutrition-focused professional learning and development for teachers; and communication between the centre and parents, as a crucial pathway to improved nutrition for children attending NZ early childhood education and care centres.

## 1. Introduction

There is strong evidence that the environments in which we live, work and play influence the development of dietary habits and food preferences [1]. Dietary habits developed in early life carry through childhood, adolescence and then into young adulthood [2,3,4], and individuals who experience obesity in childhood and adolescence are five times as likely to experience obesity in adulthood [5]. Creating healthy environments in early life, particularly in childcare settings, is a potential opportunity to address the obesity problem globally [6,7,8] and nationally [9].

Obesity prevalence was 14.9% among 4-year-olds in New Zealand in 2015/2016, a slight decrease from 16.1% in 2011/2012 [10]. While this national trends analysis indicates a decline of −0.24% each year [10], obesity levels are reported to be increasing for specific subgroups based on deprivation, ethnicity and region. Pre-schooler obesity and overweight levels were increasing in areas of higher deprivation. For tamariki (children) Māori, they increased by 0.83% per year and for Pasifika by 2.13% [10]. 

Hawke’s Bay is a region on the east coast of New Zealand where 37% of school aged children identify as Māori. There are relatively high levels of deprivation compared to national averages, with 25% of pre-schoolers in the region living in a household receiving a main government benefit [11]. Health and nutrition indicators for children in Hawke’s Bay are poor, with only 8.2% of children exclusively breastfed until the age of 6 months and 41.8% of children meeting the vegetable intake guideline of three or more servings per day across 2017–2020 (compared to the national average of 47.7%); one of the lowest intake rates in the country [12]. Pre-school obesity in Hawke’s Bay is not following the national declining pattern, with the main city of Napier in Hawke’s Bay showing the second fastest rate of increase nationally, at +1.46% per year since 2010 [10].

In New Zealand, attendance at formal childcare services (called Early Learning Services, ELS) is optional but subsidised for up to 20 h a week and so the majority of under-fives attend an ELS at least some of the week. This includes centre-based early childhood education services (ECE) including kindergartens and long day care, home-based ECE and playgroups. In Hawke’s Bay in 2021, participation in ELS was similar to national rates, with approximately two thirds of 3-year-olds and three quarters of 4-year-olds attending an ELS for 10+ h per week. Approximately one quarter of children attended ELS for 20+ h per week regionally and nationally [13]. All ELS are required to meet Ministry of Education licensing regulations that specify criteria for healthy food and drink provision, food hygiene, drinking water, supervision while eating and bottle-feeding [14]. Guidance on how to meet these standards is available through the recently revised ‘Healthy Food and Drink Guidance—Early Learning Services’, categorising foods from healthy to less healthy using a traffic light system [15] and recent guidance for reducing food-related choking [16], yet previous national ELS studies have reported ‘exceptionally low’ comprehensiveness for written nutrition policies [9].

The current paper aims to describe the food environments in early childhood education and care settings in Hawke’s Bay, regional New Zealand. Using the current guidance [15] from the Ministry of Health, this study aimed to assess the comprehensiveness of individual centre’s nutrition policies, identifying areas of strength and weakness, and report on the healthiness of menus for kai (food) provided daily in the centres. Additionally, we sought to explore common barriers and facilitators to providing a healthy food environment and to investigate educators’ perceptions of healthy-eating culture in their ELS. This survey was part of a regional initiative aiming to improve nutrition and health outcomes for children and youth in the region: Nourishing Hawke’s Bay: He wairua tō te kai (see [17] for more detail).

## 2. Materials and Methods

A cross-sectional online survey was undertaken in partnership with the Hawke’s Bay District Health Board from November 2020 to February 2021. The study population (*n* = 160) included all registered teacher- or parent-led ELS in regional Hawke’s Bay, listed in the Ministry of Education’s ELS database (December 2020). This survey excluded kōhanga reo (*n* = 57; cultural immersion centres where education and instruction are delivered in the Māori language) due to difficulties in connecting in a culturally appropriate way with them during the COVID-19 pandemic. ELS were invited via email to participate, and two reminder emails and one follow-up phone call were made to ELS if required. 

### 2.1. Questionnaire Development and Validity

The Nourishing Hawke’s Bay ELS survey was primarily informed by the most recent food environment survey of childcare services in NZ [9]. The original 65-item questionnaire was shortened to 20 items that captured the most pertinent variables influencing the healthiness of childcare food environments (see Appendix A) in order to minimise respondent burden. This 20-item survey was revised after consultation with one ELS Director, Early Childhood Education Lecturers and public health specialists from the District Health Board and the University of Auckland. Alchemer, a secure online survey software, was used to host the survey.

The questionnaire was divided into four sections: policies and programmes (whether they had a nutrition and/or drinks policy and participation in food and nutrition programmes), food provision at the ELS (daily frequency and healthiness of food provided for celebrations and used in fundraising), and ELS kaupapa/culture (perceived culture in the centre around promoting healthy eating and any barriers experienced in promoting healthy eating). These were complemented by text responses on positive change and general questions regarding demographics. Survey participants were also invited to upload or email a copy of their ELS food and nutrition policy and their food service menu.

### 2.2. Policy Analysis Using Adapted Wellness Child Care Assessment Tool

Centre policies were analysed using a shortened version of the NZ version of the Wellness Child Care Assessment Tool [9,18] (see Appendix A). This tool evaluates a centre’s food and nutrition policies based on 10 indicators across four sections (nutrition education, nutrition standards, promoting a healthy nutrition environment, and communication and evaluation). For each policy indicator, a score is given from 0 = indicator not mentioned; 1 = weak indicator; or 2 = meets/exceeds standards. From this, policy comprehensiveness and strength scores (maximum score of 10) are calculated. Comprehensiveness is calculated by counting the number of indicators rated as “1” or “2” (0 = least comprehensive, 10 = most comprehensive). It indicates how well a given policy covers the various domains of nutrition education, standards, promotion of healthy eating and communication with parents and caregivers. Policy strength (score out of 10) is calculated by counting the number of items rated as “2” (0 = weakest and 10 = strongest). It indicates whether the wording of the policy is strongly prescriptive or provides only recommendations.

Two raters independently evaluated and scored each policy for comprehensiveness and strength. Inter-rater reliability was assessed based on the percentage agreement of the scores for each indicator. We observed 79% agreement overall (110/140 indicator scores). Two indicators had less than 60% agreement: NE1 “Addresses the inclusion of nutrition education for children” and NS1 “Addresses the implementation of the Nutrition Standards for foods and beverages provided to children or food provision from home (e.g., centre provides some/all foods and/or parent provide some/all foods)” (Appendix A). All discrepant scores were discussed with a third author to reach a consensus final score.

### 2.3. Menu Analysis 

Two independent researchers categorised each item on the menus provided according to the NZ Ministry of Health Healthy Food and Drink Guidance for Early Learning Services [15] as red (unhealthy), amber or green (healthy). Green items included ‘everyday’ healthy items such as fruit, vegetables, plain milk, cereal, bread, etc. Red items were ‘occasional’ unhealthy items like muffins, pizza, sausage rolls, cake and flavoured drinks. The two raters agreed on 364 (85%) of the items before discussion, then, through discussion with a third independent rater, consensus was reached for all items. In many instances, the menus provided did not give enough detail to make an informed categorisation; for example, we were unable to tell if an item listed as ‘meat’ had had fat removed to make it eligible for the amber rating, or not, in which case it would be a red item. When there was insufficient detail provided on the menu, an assumption was made that the quantities and/or specific nutritional values of items met the criteria for the healthier ranking in the Guidelines. For example, if a ham sandwich was listed, the bread was assumed to be wholegrain, multigrain, wheatmeal or wholemeal to meet the ‘green’ standards. To give a more representative estimation of the quantity of healthy versus unhealthy items that children were being served, we also estimated the portion of the daily intake that each item accounted for. For example, if the lunch menu was a ham and cheese sandwich with a fruit platter, the sandwich was estimated to be 65% of the meal and the fruit platter 35%. The categorisation of red, amber and green was then multiplied by the weights of each item. These ‘weighted’ estimations are not reported as they were not markedly different to unweighted analysis. The nutritional status of babies’/infant menus were not analysed as the menus supplied included formula milk and vegetable or fruit puree only, nor were special dietary requirement menus.

### 2.4. Data Analyses

Statistical analyses were conducted using IBM SPSS Statistics 24. Descriptive analyses of indicators were carried out for all variables and chi-square tests where sample size allowed this. Descriptive results are reported here for all quantitative variables. The open text responses on the survey were copied into Excel for inductive thematic analysis, identifying key themes and drawing out exemplary quotes. Demographic characteristics of the centres were gathered from the Ministry of Education database as at July 2020 [19], by matching the centre ID (when provided) for four indicators: Equity Index (indicating the socio-economic status of the ELS community), type of service (Free Kindergarten, Education and Care Service, Home-based network, Playcentre) and authority (privately owned or community-based), current roll size, % European, % Māori and % other ethnicity.

This study obtained ethical approval from the Eastern Institute of Technology Research and Ethics Approvals Committee, ref 20/03.

## 3. Results

### 3.1. Characteristics of Respondents and Centres

Sixty-two survey responses were received including duplicates for two centres which were merged, creating a final database of 60 participating ELS. The response rate was 38% (60/160 centres invited). Most surveys were completed by the Centre Manager/Director/Head Teacher/Kaiwhakahaere (*n* = 45, 75%), although responses were received from three teachers/Kaiako (5%) and three parents or unpaid volunteers (3.3%) (role missing for 10, 16.7%). One third (*n* = 20) of centres provided all meals and two thirds (40) were ‘lunch-box’ services. Forty-five (75%) ELS supplied their centre identification number, so we were able to match with Ministry of Education data on the ethnicity of children enrolled, centre size, equity index, authority and type of service (Table 1).

### 3.2. Nutrition and Drink Policies in Early Learning Services

Participants were asked if their ELS had any written policies, procedures or guidelines for staff and/or parents around drinks/beverages (including messaging around water only, water as the best drink, and/or water and unflavoured milk only; covering drinks provided by the centre, brought from home, or consumed during outings or events). Fifty-two (86.7%) indicated that they did have such a document, six (10.0%) indicated they did not, and two (3.3%) did not know.

Centres were also asked whether they had any written policies, procedures or guidelines for staff and/or parents about food and/or nutrition (including guidelines about lunch box contents or practices at mealtimes, foods provided by the centre and/or brought from home). Nearly all indicated they had a food policy (*n* = 58, 96.7%), with only two centres indicating they did not (3.3%).

### 3.3. Policy Analysis Using the Adapted Wellness Child Care Assessment Tool

Of the 58 centres indicating they had nutritional policies, 14 centres provided their nutrition/food policies for analysis (24.1%). Comprehensiveness scores ranged from 3 to 8 with an average score of 5 out of 10. Strength scores ranged from 0 to 3, with only nine policies scoring above 0, and only three policies receiving a score of 3. This was mostly due to weak wording where unhealthy food and drink were not banned but discouraged. No differences between low and high equity index centres were observed in policy strength or comprehensiveness (data not shown); however, policy strength in privately-owned centres was higher than in community-based centres (2.00 vs. 0.63, respectively, not statistically tested due to the small sample size).

Regarding the individual indicators (Table 2), almost all policies provided a suitable timeframe for revising and updating their policy (92.9%, *n* = 13). Next, implementation of Nutrition Standards/Guidelines (85.7%, *n* = 12) and teachers role modelling healthy eating (85.7%, *n* = 12) were the most commonly addressed indicators. Only one centre addressed the communication of the policy to children, staff and parents. Other weak areas of policies were not addressing the provision of nutrition education training for teachers and those involved in cooking/food preparation, and the absence of nutrition standards for ‘special’ food provision (rewards, celebrations, events, and/or special occasions, and fundraising activities).

### 3.4. Food for Celebrations

ELS reported whether they provided food for special occasions and/or whether they allowed children to bring food from home for special occasions. Most centres (35, 61.4%) provided food for celebrations and allowed food to be brought from home. Food for special occasions was not allowed (not provided and not allowed from home) in five centres (8.8%); allowed only when brought from home in 11 centres (21.1%); and provided only by the centre in five (8.8%). Food brought by children for celebrations was estimated to be healthy half of the time and unhealthy the other half (ratio of green (healthy) to red (unhealthy) items of 1:1). Food provided by the centre was estimated to have a ratio of green (healthy) to red (unhealthy) items of 2.5:1.

### 3.5. Use of Food in Fundraising for Early Learning Services

Only 11 centres (18.3%) used food or beverages in fundraising (3 missing), and all of these were community-based centres. Almost a third (28.5%) of all Equity Index 5+ (highest community advantage) used food in fundraising, compared to only 7% of Equity Index 1 centres (lowest community advantage). Selling food for fundraising mostly occurred annually (*n* = 8, 13.3%). When food was used in fundraising it was generally considered to be unhealthy with the ration of green (healthy) to red (unhealthy) items estimated at 2:3. 

### 3.6. Healthiness of Food Items Provided by Early Learning Services

Ten centres provided a copy of their menu(s) for analysis (16.7%), with eighteen weekly menus for analysis. Scores for centres with multiple menus were averaged to give one score per centre. 

A total of 429 menu items were evaluated and categorised into the three nutritional categories (red, amber or green). Overall, 65% of the food items offered by ELS were categorised as ‘green’/healthy; 26% were amber (such as white buns with ham) and 8% were ‘red’/unhealthy, such as biscuits with no fruit or no wholemeal grains. 

Only four of the ten centres had menus which met the national guidelines and provided no ‘red’/unhealthy items in their menus. The proportion of ‘red’ items provided by the other centres varied from a mean of 3% to 25%, with red items such as a hotdog platter, salted caramel bliss balls, oaty choc chip muffins, devilled sausages, and banana choc chip pikelets and biscuits featuring on the menus. 

### 3.7. Barriers to Promoting or Providing Healthy Food in Childcare Centres

Table 3 presents the barriers that ELS mangers reported prevent them from providing or promoting healthy food to the children in ELS centres. Just over half (*n* = 35, 58.3%) indicated at least one barrier, with the most common barriers being lack of support from parents/whānau (30%), concerns about food-related choking (28.3%) and concerns about food intolerances or allergies (23.3%). While most centres experienced only one (37.1%) or two (25.7%) barriers, 13 centres (37.1%) reported that they experience three or more barriers. A greater proportion of privately owned centres (60%) reported three or more barriers compared to 21% of community-based centres. Almost half (46%) of low Equity Index centres experienced three or more barriers, compared to 28% of high Equity Index centres, though neither of these differences was statistically significant due to the small numbers of centres providing identification information.

### 3.8. Participation in Food and Nutrition Programmes

Almost two thirds (*n* = 37, 61.7%) of centres indicated that they participated in at least one food or nutrition programme. These programmes included both health promotion programmes such as the Healthy Heart Award [20] of the NZ Heart Foundation (*n* = 14, 23.3%) or 5+ A Day [21] (Fresh Fruit and Vegetable Charitable Trust) (*n* = 8, 13.3%); and food provision programmes such as KidsCan Charitable Trust [22] (*n* = 5, 8.3%); and the sustainability programme, Enviroschools [23] (that includes a healthy food component), was the most common programme with 18 (30%) of centres selecting this option. All centres participating in Enviroschools were community-based Kindergartens. 

### 3.9. Facilitators to Promoting Healthier Food Environments

Twenty-seven (45%) centres offered free text comments on what would enable them to create a healthier food environment in their centres. Nine key themes were identified in these comments as presented below in order of frequency mentioned. The most commonly mentioned theme was the need for more food and nutrition education for parents (8/27 responses). Centres suggested, for example, workshops on healthy lunch boxes or introducing nutrition messages into parent newsletters. This is consistent with the biggest perceived barrier to promoting healthy nutrition being a lack of support from parents/whānau. Several participants acknowledged that families’ linguistic, cultural and socio-economic settings can make it more difficult to communicate and promote healthy eating messages. 


*“More parent education - not always easy as we have a diverse community and language can be a barrier”. (Centre Manager of high advantage community-based kindergarten)*


More buy-in/support from staff and more professional learning and development for staff were also mentioned in several comments. A number of centres indicated that not all staff members and teachers supported the restriction of ‘treat’ food for special occasions and that more training was required so that busy teaching teams did not feel overwhelmed by having to add ‘yet another’ aspect to the curriculum. Several comments mentioned the need for more participation in health-promoting programmes like the Healthy Heart Programme, or Sport Hawke’s Bay: 


*“We worked really hard for two years changing culture and aligned it to physical active play with SportsHB. [We need] lots of communication within staff team & parents, and to maintain consistency of staff, also [this is a] milestone in our strategic plan under sustainability”. (Centre Manager of low advantage community-based kindergarten)*


Several centres suggested that direct funding to provide healthy kai (food) was necessary, either because working within the existing budget was difficult, or so that they could stop food being brought from home and provide all snacks and meals for tamariki (children) to ensure it was of a high nutritional quality. Some centres mentioned that vegetable gardens would enable them to provide healthier food to their children.

Another theme was several centres mentioning that improvements to, or fewer restrictions in the current Ministry of Education (MoE) guidelines for food provision in ELS were necessary:


*“We grow many vegetables, and our whānau (families) regularly bring in fresh fruit and vegetables they have grown in their gardens. The new [choking] regulations put a fear amongst whānau, children and teachers about a natural way of eating… If they do not learn in these early years how to chew, masticate and swallow there may be more problems in school when they are expected to eat unsupervised”. (Centre Manager of low advantage community-based kindergarten)*



*“It is with much regret that our large grapevine, our apple tree and our vege gardens are now deemed unsafe for our children to enjoy independently. We now have to net it all up, robbing them of something precious, robbing them of growing and eating their own in their learning environment. A healthier food environment DID mean spending the past 4 years growing eating and enjoying fruits and vegetables in their natural form in our environment. It is now controlled, and served in a form that is foreign to us all”. (Centre Manager of low advantage community-based kindergarten)*


Finally, ELS managers saw a need for nutrition to be embedded in the curriculum, healthier options for celebration food and recipe ideas, particularly for nutritious recipes that met the new national anti-choking guidelines.

### 3.10. Culture around Healthy Eating

Participants were asked if the culture around healthy eating in their centres was ‘Very Strong’ (policies in place, strong healthy food practices, staff and parents strongly support kaupapa (topic) of healthy food); ‘Strong’; ‘Medium’ (some policies and practices support healthy food, mixed support from staff and parents for the kaupapa of healthy food); ‘Weak’; to ‘Very weak’ (no policy, considerable unhealthy foods provided, healthy eating is a low priority for staff and parents). The majority indicated that the healthy eating culture in their ELS was Strong or Very Strong: 16 centres (26.7%) felt the culture was Very Strong, 28 (46.7%) Strong, and 10 (18.5%) indicated it was Medium; six (10.0%) were missing. 

We asked participants to further qualify the culture of healthy eating at their centres by indicating their agreement (5-point Likert scale from Strongly agree to Strongly disagree) with a number of positive statements around strong healthy eating cultures in centres. Rates of agreement are shown in the following Table 4.

There were no statistically significant differences in overall culture scores between privately owned and community-based centres, or low and high equity index centres (*p* = ns for both). However, two trends were observed with community-based centres being less likely to provide and update nutrition training for staff (32.1% strongly agree or agree), compared to privately owned centres (64.7%); and community ELS were more likely to communicate with parents and whānau about healthy eating (81.5% strongly agree or agree vs. 64.7% in privately owned).

These results were illustrated in an infographic which was sent to all regional Early Learning Services (participating and non-participating), as well as stakeholders, as important feedback on policy and practice (Appendix A).

## 4. Discussion

The current study provides a snapshot of policies and practices in ELS in Hawke’s Bay, New Zealand. Almost all centres reported having a policy or guidelines around food/nutrition (97%) and drinks/beverages (87%), higher than previous national figures, e.g., Pledger et al. [24] reporting that 88% had a food and nutrition policy; or regionally in Auckland and surrounding regions reporting that 82% of ELS had a written healthy food, nutrition or wellness/hauora policy [9]. Australian data from 2015 indicated that only 58% of centre-based childcare services had written healthy eating and physical activity policies [25]. The higher number of centres we observed may reflect changes in the national policy context with more support for centres and a new Healthy Food and Drink Guidance for Early Learning Services released in 2020 [15]. 

While policy existence in ELS was high, most policies made references to the national food and nutrition guidelines, but rarely mandated or monitored their implementation. In addition, our analysis of policy strengths and weaknesses identified that only one policy addressed the communication of the policy to children, staff and parents. As parents are primary decision-makers regarding food for their children, a lack of communication between the ELS and parents posits a challenge. Providers may sometimes lack the confidence [26,27] or feel uncomfortable discussing nutrition and dietary behaviours with parents [28], and according to a government-led review in 2016, do not want to be seen as the ‘food police’ [29]. However, strategies targeting parents are necessary to improve dietary intake in childcare [30] and knowledge gaps can be overcome through educational resources and meetings. The lack of reported support from parents/whānau (30.0%) mirrors the large group of centres citing nutrition education for parents as the most common facilitator to promote a healthier environment, also reported by the ELS sector in other regions [9]. 

A notable strength of policies analyses and an improvement since a similar NZ ‘Kai Time’ survey in 2014 [9], was the increase in policies addressing teachers’ role modelling of healthy eating. Educator–child modelling affords the opportunity to provide nutrition education to children [31]. However, it is important for future research to evaluate the implementation of this policy indicator, as the literature shows that staff are often unable to effectively model due to feeding and responding to a large number of children simultaneously [26,28].

Unlike the policies evaluated in Kai Time 2014, our survey identified a lack of provision at the policy level of nutrition education for teachers. This finding was supported by managers reporting that a lack of training was a barrier to healthy food promotion and provision, and by the qualitative comments regarding lack of training. Nutrition education has also been shown to increase knowledge, staff efficacy and in turn improve nutrition environments [32,33], and is often offered in NZ as part of involvement in health-promotion programmes. Two thirds of centres in our study participated in a food or nutrition programme and some reported that this participation was key to promoting a healthier food environment. Successful implementation of dietary guidelines in childcare services can be dependent on training to upskill staff, alongside acquiring new foods and cooking apparatus which increased expenses [34]. It is important to evaluate the type of nutrition education and training offered to NZ childcare educators as the last stocktake of this in NZ was in 2016 [35]. 

Most centres (67%) in this research were providing food to children at least some of the time. Around a third provided all food (minimum of lunch and two snacks), similar to the findings of Kai Time (33.5%) [36], and a majority of the remaining ELS provided snacks and celebration foods. Of the 10 centres in our study that provided menus for analysis, menus from only four centres met the national guidelines, containing no unhealthy (‘red’) items. This is similar to previous research in NZ [9]. International research has also noted poor adherence to dietary guidelines [37,38] and so this is another important area for improvement. A recent consolidation of systematic reviews has highlighted the positive association between healthy food availability and improved dietary intake while in childcare [39]. 

One particular area of food provision where food was less likely to be healthy was for celebration food. A small-scale (*n* = 12 participants) Canadian study reports that unhealthy food was often used to celebrate special occasions (e.g., Christmas, Mother’s Day) and teachers sometimes rewarded healthy eating with less nutritious food [28]. We noted the use of marshmallows for children’s birthdays in one ELS, a practice which may undermine other positive food environment initiatives and send mixed messages to children. However, overall, ELS staff estimated that celebration food was 2.5 times more likely to be healthy when provided by the centre than when brought from home. Although previous NZ research found cakes, biscuits, pies and sausage rolls to be the most popular items used in celebrations [9], this healthier approach from centres in our regional results might be the result of the new national guidelines [15] and resources such as the ‘Healthy Celebrations’ book from the Heart Foundation, offering recipes and ideas for healthier alternatives at celebrations [40]. In comparison with Gerritsen et al. [9], where one third of their centres used food in fundraising, only 18.3% of participating centres used food or beverages for fundraising. Although the predominant use of unhealthy items in fundraising is consistent with the results from earlier research, this is a notable reduction in the number of centres utilising foods/beverages for fundraising [9]. 

In 2020, just prior to our survey, the Licensing Criteria for Early Childhood Education Care Centres 2008 was updated (officially released in April 2021) to include more detailed guidance for the supervision of children while eating to reduce the risk of food-related choking [14]. Additionally, foods that pose a high risk must now be prepared according to Ministry of Health guidelines or not served at all [15]. These new criteria have been interpreted by centres as a restriction in the foods they can offer and requiring changes to the methods in which they are allowed to prepare them, as well as requiring ELS to keep fruit and vegetable gardens off limits from children. As a major unintended consequence of a well-intentioned policy, 28% of our ELS perceived the new guidelines to be a barrier to providing healthy foods. 

The main limitation of this study is the size of the sample, though with a 38% response rate, there is good coverage of different types of centres and a good range of equity indices. Potential social desirability bias should be taken into account for all questions, particularly with regard to culture of healthy eating. As participation was voluntary, and to minimise inaccurate data and dropout rates [41], the survey design forfeited forced answering. However, this resulted in a number of questions with missing data. Additionally, only a handful of centres (16.7%, *n* = 10) provided a copy of their menu for analysis. With limited information on ingredients and portion sizes, several ‘healthier’ assumptions were made when analysing food service menus. However, this still serves to provide a good snapshot of current food provision menus. In addition, this study did not evaluate food provided by parents/whānau (in 27.0% of centres), an important factor influencing children’s nutrition while in childcare. Future research would benefit from third party observations, to validate that the menus reflected the actual food served and eaten by children. However, this survey combines an investigation of practices, policies and menus in a short survey in this regional population with high deprivation and poor health indicators, giving a unique snapshot of perceptions and practices, just following the introduction of new national guidance on food in childcare settings. 

## 5. Conclusions

Guided by previous research and using our newly adapted tools, updated and shorter versions of the Kai Time Survey questionnaire and the WellCCAT-NZ policy analysis instrument (Appendix A), this is the first study to assess the healthiness of food environments in ELS in Hawke’s Bay. We have attempted to understand the contribution and importance of a culture/kaupapa of healthy eating in establishing healthy ELS food environments, and responses show that while most ELS felt they had a strong culture of healthy eating, this was not always reflected in their policies and practices. Most menus do not adhere to current guidelines regarding ‘red’ unhealthy items; fundraising activities utilise unhealthy foods and beverages, and many centres report barriers to promoting or providing healthy food. In communities such as Hawkes’ Bay, where pre-schooler obesity has not followed the national pattern of declining rates and where there is a large Māori and Pacific population with high health disparities, increased equity-focussed efforts in nutrition policy development and interventions are necessary. The Early Learning Services setting, attended by a majority of children and potentially the area where children are first exposed to different foods and ways of eating, is an ideal setting for health-promoting nutrition. We therefore highlight three main foci for improvement of Early Learning Services food environments: the need for more widespread policies regarding water/milk-only and healthy food used in celebrations and fundraising; the need for increased professional learning and development in health-promoting food environments for teachers; and a greater emphasis on communication between the centre and parents, as a crucial pathway to improved nutrition for children attending early childhood education and care.

## Figures and Tables

**Table 1 ijerph-19-04709-t001:** Participant and Early Learning Service characteristics.

Characteristic of Centre	N (%/45)
Type of Service *	
Education and Care service	25 (55.6)
Free kindergarten	17 (37.8)
Home-based network	1 (2.2)
Playcentre	2 (4.4)
Authority	
Community-based	28 (62.2)
Privately owned	17 (37.8)
Equity Index **	
Low equity index/low advantage	16 (35.6)
High equity index/high advantage	29 (64.4)
Average roll size (range)	43.5 (9–96)
Ethnicity of children enrolled at centre	
NZ European (range)	51.1% (0–96)
Māori (range)	36.6% (0–100)
Other ethnicity (range)	12.3% (0–43)
Provision of kai (food) and types of meals provided by centre (made on site)	
Centre provides some food/meals	24 (40.0)
All food provided by centre	18 (30.0)
All food provided by home Missing	16 (26.7) 2 (3.3)
Type of food provided (of 42 centres providing food) Breakfast	8 (19.0)
Morning snack	40 (95.2)
Lunch	27 (64.3)
Afternoon snack	32 (76.2)
Late snack	12 (28.6)

* For more detailed definitions of the types of services, see https://www.educationcounts.govt.nz/directories/early-childhood-services (accessed on 15 November 2021); ** low = equity indices 1–3, high = equity indices 4 and 5 +; ELS in New Zealand can be teacher-led (kindergartens and education, home-based education and care, care services or correspondence school), parent-led (playcentres or playgroups, including language immersion playgroups), or whānau-led (kōhanga reo Māori immersion) https://parents.education.govt.nz/early-learning/early-childhood-education/different-kinds-of-early-childhood-education/ (accessed on 28 March 2022).

**Table 2 ijerph-19-04709-t002:** Policy analysis indicators across nutrition education, nutrition standards, promoting healthy food, and food and nutrition communication (shortened WellCCAT-NZ *).

Indicator Name	Indicator Description	Average Score (0–2)
Nutrition Education		
NE1	Addresses the inclusion of nutrition education for children.	0.71
NE2	Addresses the provision of nutrition education training for teachers and those involved in cooking/food preparation.	0.36
Nutrition Standards		
NS1	Addresses the implementation of the Nutrition Standards for foods and beverages provided to children or food provision from home (e.g., centre provides some/all foods and/or parent provide some/all foods).	1.07
NS2	Addresses implementation of the Nutrition Standards for rewards, celebrations, events, and/or special occasions, and fundraising activities.	0.36
NS3	States that beverage provision is milk and water only (no sugary drinks at any time).	0.43
Promotion of Healthy Food and Nutrition Environment		
NP1	Encourages teachers to be role models for healthy eating (e.g., sitting with children during meals, assisting children to gauge fullness) including staff consumption of foods and/or beverages meeting the Nutrition Standards.	1.07
NP2	Addresses specific course of action when food from home does not meet nutritional standards.	0.42
Food and nutrition communication and evaluation		
CE1	Addresses the communication of the centre food and nutrition policy to children, staff and parents.	0.07
CE2	Addresses the provision of nutrition information for parents.	0.57
CE3	Specifies a suitable timeframe for revising and updating the centre food and nutrition policy.	1.14

* Wellness Child Care Assessment Tool [18] adapted for use in NZ and to align with NZ nutrition guidelines [9].

**Table 3 ijerph-19-04709-t003:** Participation in food programmes and barriers to creating healthy food environments in Early Learning Services in Hawke’s Bay.

	N (%)
**Barriers (Multiple Possible)**	
No barriers experienced	25 (41.7)
Lack of support from parents/whānau	18 (30.0)
Concerns about food-related choking	17 (28.3)
Concerns about food intolerances or allergies	14 (23.3)
Insufficient funds	12 (20.0)
Lack of staff training on nutrition education	7 (11.7)
Lack of training for cook/food service staff	6 (10.0)
Requirements around food safety	6 (10.0)
Lack of support from teachers and/or staff	3 (5.0)
Lack of support from cook/food service staff	3 (5.0)
Sales of unhealthy foods as fundraisers	3 (5.0)
Inadequate food preparation or storage facilities	3 (5.0)
Lack of resources/information on health food for children/tamariki	3 (5.0)
Lack of support from administration or management	2 (3.3)
**Participation in food and nutrition programmes**	
Enviroschools/Te Aho Tu Ra	18 (30.0)
Healthy Heart Award (Heart Foundation)	14 (23.3)
5+ A Day (Fresh Fruit and Vegetable Charitable Trust)	8 (13.3)
KidsCan Charitable Trust	5 (8.3)
Other programmes (Nourish, own vegetable gardens)	6 (10.0)

**Table 4 ijerph-19-04709-t004:** Culture around nutrition in Early Learning Services in Hawke’s Bay.

Nutrition Culture in Centre Statements	Mean Score ¥ [95% CI]	Strongly Agree/Agree * N (%)	Disagree/Strongly Disagree * N (%)
Centre management and staff share a strong collective vision around hauora/health	4.30 [4.06–4.55]	49 (81.7)	3 (5.0)
Nutrition and healthy eating are highly prioritized at our centre (incorporated into service policy/vision/goals)	4.27 [4.06–4.48]	48 (80.0)	1 (1.7)
Staff consistently act as role models for healthy eating (teachers sit down and eat with tamariki (children), and eat healthy food in front of children)	3.96 [3.71–4.22]	43 (71.7)	6 (10.0)
We frequently communicate with parents and whānau about nutrition and healthy eating (e.g., through enrolment information, newsletters, website/Facebook, posters, app)	3.89 [3.63–4.16]	39 (65.0)	5 (8.3)
Our centre and parents/whānau share a strong collective vision around hauora/health	3.63 [3.41–3.84]	34 (56.7)	3 (5.0)
Nutrition training is provided and regularly updated for all staff (including cooks and food service staff)	3.25 [2.98–3.52]	25 (41.7)	12 (20.0)

* Combines two categories of strongly agree and agree; combines strongly disagree and disagree; remaining category: neither agree nor disagree. ¥ Likert scale from 1 strongly disagree to 5 strongly agree.

## Data Availability

The full data set is available on request from the authors.

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
