# Peer review of "Healthy Food Environments in Early Learning Services: An Analysis of Manager Survey Responses, Menus and Policies in Regional New Zealand Early Childhood Education and Care Centres"

_ijerph, 2022, doi:10.3390/ijerph19084709_

Round 1
Reviewer 1 Report
A great study and thank you for the opportunity to review. Please note some thoughts below;
- the abstract would benefit from more detail of analysis method
- Australian context - Childcare are now referred to Early Childhood Education and Care as it supports the narrative beyond the traditional "Baby Sitting" service associated with childcare. However I would still include the word "childcare' in key words to support article searches. NZ - Refer to the services as ELS to support this narrative, where possible.
- Include more information in the background to support the elements of your aim that is, evidence demonstrating elements of comprehensive of nutrition policies for inclusion, definition of healthy culture (for international context) and expectations.
- include total number of non-teacher led services other than the 57 excluded.
- What validity measure were used for the revised questionnaire - content and face validity...?
- on occasion first person was used ("we" and "our")- please remedy to improve scientific tone of manuscript
- Line 232 - As a result of the reporting on proportion, which I am assuming was by frequency proportion rather than weight. I would suggest removing the requirement for weight in the methods as it seems unnecessary (line 139).
- Line 172 - Table 1 reports on Education and Care Service is this a type of ELS, if so this needs more detail in the background to avoid confusion. Line 56 - add detail here.
- You need to be consistent with tool alignment eg if tool classifies popcorn as red then your data should reflect this.
- line 148-149 These ‘weighted’ estimations 148 are not reported as they were not markedly different to unweighted analysis.- requires further explanation
- If babies were not included, how did you estimate provision? as you haven't specified an age group - Line 149-150 "Babies’ menus were not analysed if they only included formula milk and vegetable or fruit puree, nor were special dietary requirement menus.", can you please provide justification.
- review table 1 as how this is presented doesn't represent the components that need to total 100% very easily. For example the type of service either community/privately owned should total 100%
- It would be good to see the breakdown of other cultures rather than just European, to identify possible support for services in cultural competence.
- The context of each of the service environments need further explanation as some are more for full day care and curriculum provision and others are focused on curriculum only, and others are focused more on care, and some only provide only short term care. These distinctions are important when reviewing policy inclusions as the focus and intent of the policy will reflect the policy inclusions.
- There are some inconsistencies in language - are they children or students?
- Table 2 - are you able to identify the elements for each of the four areas and list the indicators under each of these elements?
- Line 242 - mean/median
- table 3 - first column align left
- There is benefit in looking at the services open ended comments by service type considering the uniqueness of each one. There could be a danger in making recommendations across the different services - it would be good to see the barriers by service type even if included as supp materials as I appreciate this reduces the sample size for reporting.
- Reference the programs under 3.9.
- Line 266 - was there nothing worth reporting for the other programs?
- Can you share the data as to how many services were lunchbox services compared to meal provision services in your results as a demographic?
- Your quotes only represent Centre managers, if you were to include quotes from teachers etc, you would need to further de-identify due to low response rate.
- Line 294 is great evidence
- Line 335 highlights a contradiction in that private are more like to provide PD yet they have poorer quality menus, does this mean the PD quality needs reviewing (a discussion point)?
- Limitation for questions especially culture based questions - social desirability bias
- Line 392-394 needs to be more evident in descriptives
- Line 395- 401 needs to be reworded to improve clarity of meaning
- Line 414 - 18% is quite a lot, beware of subjectivity in comments where you have not bench marked against other research
- Provide ECEC licensing criteria information in the background including the timing and key elements of its release
- Line 423 - what was the old criteria?
- Line 426 - this is a shame and likely results from a lack of guidance and understanding of possible management strategies in the policy department.
- be consistent with language ELS/ECEC/childcare??
- Line 446 - did you create all the tools, avoid sweeping generalisations. Perhaps a more accurate statement would be adapted tools which require further validation
- Line 453, there was no evidence of this in your data - "weak"
- The interchange of terminology between preschool and childcare setting - does not demonstrate your understanding of the unique differences between these settings and the differences which will be required to support these settings. I would suggest fleshing this out earlier in the background and in the breakdown of the findings which will link well to the foci/recommendations.
- The research study was gaining a snapshot of policy, menus and managers survey responses, your recommendations need to reflect each of the findings for these elements and how stakeholders could work together to build capacity for system change. For example - there is weak policy so perhaps development of an exemplar policy to be share with implementation agencies and programs could improve policy quality. Professional development could be linked back to specific elements in policy and mangers could undertake training to facilitate conversations with their local government or licensing body to develop practical solutions for inclusion of vegetable gardens to overcome the barriers.
This is a very important piece of work - you have loosely identified the gaps some areas need further development. I encourage you to update the manuscript with the suggested changes and send in for review.
Reviewer 2 Report
Dear authors, I had read with interest your contribution, which is a good one to improve the food environment in schools in a multicultural one. But there are just two comments which could improve your report. • Keywords. To improve the indexation of your work please consider adding the following MeSH word: Child Welfare; Child Day Care Centers. • Materials and methods. Due to the use of both methodology (qualitative and quantitative) please add more details about thematic analysis. • Results. Could you add more quotes about the facilitators and barriers to a healthier environment?
Round 2
Reviewer 1 Report
I am happy for the editors to review.
- Please note, on first glance I can see the incorporation of ECEC and then a switch to ELS.
In Australia ECEC incorporate Long day care, family day care and outside school hours care. In NZ it looks like ELS include Education and Care Service (is this Long day care/centre based care) , free kindergarten, homebased network and playcentre, can the NZ context be provided in the intro (line 59), which should be referenced and then results and discussion accurately reflect the correct term. I can see a definition (incorrectly referenced) in the footnote of table 1 however on review of this link it does not provide the information which supports the definition, this will need to be reviewed and updated.
I am unfamiliar with NZ structure and have found the information below and I strongly encourage reporting to accurately reflect legislative and regulatory definitions and will defer this to the authors to ensure the correct definitions are captured and then consistently used throughout the paper. Please see below;
The regulations refer to these services as ECS (https://www.legislation.govt.nz/regulation/public/2008/0204/latest/DLM1412506.html)
- There seems to be a different meaning for ECEC and ECES
https://www.legislation.govt.nz/act/public/2020/0038/latest/LMS171311.html#LMS171311
- table 2 – second column should align left
